# Modular Interorganizational Network Governance: A Conceptual Framework for Addressing Complex Social Problems

**Srivardhini K. Jha [1,*]**, **E. Richard Gold [2]** and **Laurette Dubé [3]**

1 Indian Institute of Management Bangalore, Entrepreneurship Area, F202, Bannerghatta Road, Bangalore 560076, India

2 Faculty of Law, McGill University, 3644 Peel Street, Room 408, Montreal, QC H3A 1W9, Canada; richard.gold2@mcgill.ca

3 McGill Centre for the Convergence of Health and Economics, McGill University, 3430 McTavish St., Montréal, QC H3A 1X9, Canada; laurette.dube@mcgill.ca

* Correspondence: srivardhini.jha@iimb.ac.in

**Abstract:** We develop a conceptual governance framework to guide creating and managing a modular interorganizational network to address complex social problems. Drawing on theoretical foundations in modularity and interorganizational networks, we propose that modularizing complex social problems is a dialectic, emergent process that blends a convener-led network formation with a consultative problem definition and solution design. We also posit that social systems are imperfectly modular and need purposefully designed interface governance to integrate the modules. Finally, we advance how leveraging modularity may simultaneously advance the interests of participating actors and deliver societal value. Together, the propositions advance a governance framework for a modular, multi-actor adaptive system suited to tackle the scale, diversity, and dynamics of complex social problems.

**Keywords:** social problems; modularity; network governance; value creation; interdependence; convergent innovation

## 1. Introduction

The most significant challenges to overall well-being in the 21st century—poverty, malnutrition, environmental degradation, climate change, and universal health care—remain intractable because they comprise a set of interconnected, smaller problems that cross boundaries between industrial (e.g., agriculture, food, transportation, and healthcare) and societal (e.g., for-profit, non-profit and government) sectors [1,2]. As a result, they are beyond the mandate and capability of individual organizations. Labeled 'grand challenges' [1,2] or 'wicked problems' [3,4], these problems are subjectively constructed and interpreted by different actors, pose uncertainty in terms of cause–effect relationships, and exhibit dynamic complexity, that is, they evolve over time [2,5]. Several proposed approaches attempt to tackle these problems: robust action [2]; large systems change [6]; collective impact [7–9]; and convergent innovation [10–12].

Each of these approaches hinges on operating through an interorganizational network that transcends sectors, geographies, and jurisdictions to draw on actors' distributed and complementary capabilities to address the various facets of the problem. Together, they have advanced several strategies and best practices to orchestrate collective action [7,8] and achieve superior social outcomes [2]. However, two critical gaps exist. First, multiple approaches moot the idea of seeding a portfolio of loosely coupled modules [11], or working groups [8], that address a specific sub-problem within the broader problem domain wherein actors who can contribute to and benefit from the solution are involved. Unfortunately, there has not been scholarly attention on the important process of breaking down the

complex problems into modules, the smooth functioning of the individual modules, and how they come together to form a complete solution.

Second, although there is broad agreement that complex social problems need orchestrated action from a cross-section of societal actors, limited attention has been paid to the different notions and expectations of the value of the participating actors. There persists a well-entrenched divide between the goals and mandates of for-profit, non-profit and governmental sectors. While the for-profit sector is primarily concerned with wealth creation, the non-profit and government sectors give primacy to social development and environmental conservation [10,13–16]. Such a divide implies that any approach that proposes to assemble an array of actors cutting across societal sectors needs to ensure that it creates value for each of the participating actors and to the society as a whole [17–19].

Addressing these gaps, we explore in the context of complex social problems the issue of governance, i.e., the set of coordinating, monitoring, and value-creating mechanisms that enable organizations and their collaborative relationships to survive and thrive [20]. Specifically, we ask the following:

> *How to design the governance framework for a multi-actor, modular system to address complex social problems that simultaneously advances the interests of participating actors and creates societal value?*

We draw on two bodies of literature to advance the conceptual governance framework. First, we leverage the literature on modularity. Modularity is a dominant design principle in technical systems and provides a way to break down complex problems into largely independent yet loosely coupled manageable components that together create individual and system-level outcomes [21,22]. Each component, or module, addresses a part of the overarching problem and provides a mechanism to manage the converging and conflicting interests of the spectrum of actors. However, the assumptions of modularity with respect to hierarchically governed modules and complete independence between modules do not seamlessly extend into the social sphere. In order to overcome this limitation and extend the principle of modularity to social systems, we weave this together with the literature on interorganizational network governance [23–27]. This literature allows us to view each module as a collaborative interorganizational network [28] that may have a range of governance mechanisms based on the characteristics of the network. It also allows us to theorize on the nature of interface governance needed for social systems where modules may not be completely independent. Together, these two streams of literature provide the theoretical backbone upon which to conceptualize the governance of a quasi-modular network: the set of coordinating and monitoring mechanisms that enable the orchestration and integration of modules that address a complex societal problem.

We argue that modularization is an emergent, dialectic process that blends a convenor-led network formation that catalyzes system-level change, with problem definition and solution design emerging through dialogue and the synthesis of participating actors' views, interests, knowledge and capabilities. We also posit that social systems are imperfectly modular, which means that some interdependencies will remain between modules, even after modularization. The type of interdependence has a bearing on the 'modular interface governance.' Finally, we discuss the notion of value for participating actors and value for broader society and how modular network governance might balance the two types of value. Taken together, our propositions advance a governance framework for a modular, multi-actor adaptive system suited to tackle the contextual diversity and dynamics of complex social problems. The paper breaks new ground by extending modularity into the social domain and lays a foundation for future research inquiry. It also provides an actionable framework for academics, practitioners, and policy makers to experiment with and refine.

The remainder of the paper is organized as follows. We first review the relevant streams of literature, drawing out the key insights that will inform the development of our governance framework and identifying the critical gaps that exist. We then advance the framework and conclude with a discussion on how this contributes to theory and practice.

## 2. Literature Review

### 2.1. Modularity and Social Problems

Modularity is a design principle that separates systems into manageable components, known as modules, in such a way as to maximize both interdependences within individual modules and independence between modules. Splitting systems into modules ensures individual and system-level outcomes [21,22,29,30]. Indeed, each module is a tightly knit structure that abstracts a bundle of interdependencies, requiring constant communication and coordination between the participating elements [29]. Ideally, modules are largely independent of one another, offering the flexibility to modify individual modules without affecting the entire system, provided that each module adheres to a predefined set of interface standards [29]. Therefore, designing a modular system involves defining an architecture consisting of modules that constitute the system and their roles and the interfaces between the modules [29,30].

The general set of principles propounded by modularity can be applied to manage large problems, whether in relation to product development, production systems, or organizations [31]. Furthermore, one of the big advantages of a modular structure is in leveraging this structure to facilitate cooperation among a disparate set of actors in reaching the system-level goal by isolating competitive and conflicting elements into separate modules and ensuring the protection of intellectual property and fair appropriation of value by the participating organizations [21]. Finally, the notion of modular interface and its definition provides a mechanism to integrate the modules into a single solution system.

Pioneering in the technology sector [21], the primary mechanism to govern modular systems is through the market. The suitability of market governance stems from the fundamental assumption that once module interfaces are defined, there is minimal interdependence between them and, hence, little need for communication and coordination other than through price. This assumption implies the possibility of developing an individual module by multiple firms in a competitive market and transacted with downstream businesses or end consumers. Thus, an initial effort is made to define the interface standards after which market mechanisms take over and ensure the system's functioning. However, this mode of governance can only work in a relatively static environment in which interface standards change infrequently; else, the cost of operating through markets would be prohibitively high [32].

A proposed societal scale solution utilizes modularity as a tool to address complex social problems by breaking them into a portfolio of smaller, goal-oriented modules [11]. However, we need to address two issues when attempting to transplant the design and governance of modular technical systems to a modular system for these complex problems. First, we need to explore the question of if and how societal problems can be modularized. Further, the nature of modular interface needs to be defined. In technical systems, interfaces are technical standards (e.g., USB). What is the equivalent in social systems? A codified set of standards and practices combined with auditing protocols has been used to certify and unify sustainability practices across geographies in various industries [33]. Further attention is required when considering that a similar approach might fall short when the modules that need to be integrated are very different from one another, i.e., where module diversity is high.

Second, the literature on the governance of modular systems has only focused on systemic governance, i.e., formulating the rules around decomposing the system and defining the interfaces needed to integrate them. Treating the individual modules as black boxes and assuming, as the literature does, that their governance has no bearing on the system so long as each module adheres to the interface specifications is a limitation. This assumption stems from the understanding that, at the modular level, there is an implicit one-to-one correspondence between the target activity/program/technology and the business that puts it on the market. In other words, it is assumed that a module is completely under the control of a single organization, and therefore, its governance is the prerogative of that organization. However, this assumption is not applicable for systems

aiming to address complex societal problems because, in these systems, many individual modules might involve a collaborative effort between multiple organizations [11]. In other words, modules are likely to be interorganizational networks, and the governance needs to be carefully designed as a function of their respective compositions and goals. Thus, we turn our attention to the literature on the governance of interorganizational networks.

### 2.2. Governance of Interorganizational Networks

A network is a set of nodes and the set of ties representing some relationship, or lack of relationship, between the nodes. The nodes are referred to as actors or entities and can be individuals, work units, or organizations [34]. An interorganizational network is one in which each node is an independent organization, and the ties represent the relationship between those organizations. Governance is the set of coordinating and monitoring mechanisms that enable organizations and the collaborative networks they form to survive and thrive [20]. In other words, governance resolves the fundamental issues around initiating, adapting, coordinating, and safeguarding exchanges between network actors [35]. It does so by articulating which network actors drive key decisions for the collaboration and the mechanisms (formal, informal, or a combination) they employ to arrive at those decisions.

Scholarly study of interorganizational networks and their governance has proliferated substantially over the last few decades, recognizing that organizations often operate outside their hierarchies and markets in various collaborative relationships. As a result, the literature has developed along two distinct streams, each focused on a different level of analysis. The first is grounded in strategic management and concerns the networks of a focal organization and its impact on organizational outcomes [25,34,36]. The second stream is grounded in public administration and cross-sectoral social partnerships and looks at the overall network and outcomes at the network level [20,37–42]. This paper focuses on the latter, which we refer to as 'whole networks.'

### 2.3. Whole Network Governance

Research on 'whole networks' focuses on the network itself [41,43]. This focus has come from recognizing that purposefully formed, goal-driven interorganizational networks provide a mechanism to achieve ends that no organization can unilaterally achieve [37,41,44]. An example is delivering a public service through a network of community-based and private organizations [37]. The governance mechanisms are designed to overcome structural and relational challenges in these networks to achieve network-level outcomes.

Provan and Kenis [26] put forth two types of whole network governance mechanisms. In the first, the network is jointly governed by all or a significant subset of the participating actors and is called *shared governance*. Shared governance is a decentralized form of governance characterized by dense interactions between participating organizations. Effective governance depends on the involvement and commitment of all (or mostly all) network participants, who manage internal network relationships and external ones. Thus, there is no formal administrative entity, and the power equation in the network is more or less symmetrical.

The second form of network governance is *brokered governance*. This is a centralized form of governance in which network level decisions and orchestration are carried out by a broker. A broker is an entity that can connect actors who are otherwise not connected [23,27,45]. Brokers bridge structural holes, which are the relational chasm separating groups of actors who have their own distinctive ways of thinking and operating [23]. By virtue of their structural position, brokers can access information from different groups, and synthesize and generate ideas at the intersection of different bodies of knowledge. This fortunate position gives them the leverage to act as the anchor, exerting influence and facilitating interaction between other actors [23]. Brokered governance may be administered by a lead, participating organization, or a separate network administration

organization (NAO) [26]. In lead organization–governed networks, the lead organization is typically a key participant, wielding a disproportionately high degree of power. All network activities and key decisions are coordinated by and through this organization [26]. It is a centralized form of governance where the decision making and coordination are top-down. By contrast, in the NAO form of governance, a separate non-participating entity is responsible for facilitating the decisions and activities of the network. This entity is often created/appointed by the participants and collaborates with them to govern the network. The decision making is representative of all members, even though the coordination function is centralized.

The decision of which governance mechanism is most suitable for a given interorganizational network depends on three factors. The first is the structural characteristics of the network [26], which comprises factors such as the size of the network, the level of interdependence among network actors, and the level of external-facing activity of the network. As these factors become dominant, the need to coordinate also becomes greater, moving the network toward a brokered governance mechanism, such as the NAO [26]. The second factor is the relational characteristics of the network [26]. These relational characteristics include the density of trust in the network [26,42,46] and power distribution among network actors [20]. Governance tends to be shared if trust is high and power imbalances can be mutually managed. Otherwise, an NAO or lead organization model is likely to prevail. Finally, the third determinant of the mode of governance is the strategic purpose of the network. When the collaboration is strategically important to the participating actors, the frequency and intensity of interactions and resources exchanged increase. In such cases where the governance function is intensive, the network is likely to adopt the more formal NAO model of governance [26]. This literature lays a strong foundation for designing the appropriate governance based on the characteristics of the network under consideration. Despite this, a critical gap exists in that the cross-sector network governance literature does not give due attention to the value dynamics of the network.

Networks can create societal value, which is the value that accrues to society at large due to the partnership [46–48]. They can also create partner value, i.e., value that accrues to the actors involved in the partnership by advancing their interests, be these financial or otherwise. Austin and Seitanidi [49,50] identify four types of partner value: associational value, transferred resource value, interaction value, and synergistic value. Associational value is the benefit that accrues simply from having a collaborative relationship (e.g., enhanced reputation and credibility). Transferred resource value derives from receiving a resource (e.g., cash, equipment) from a partner. Interactive value is the intangible benefit derived from working together, such as trust, knowledge, and problem-solving techniques. Finally, synergistic value is in many ways the *raison d'etre* for collaborations, which allows each partner to achieve more by partnering than they could achieve independently.

In sum, partner value can take on a variety of forms, some of which create direct economic benefit to the partnering organizations. In contrast, others contribute to improving reputation and perception, facilitating learning and innovation, or achieving one's institutional (often socially oriented) mandate. In principle, the goal of the network is to create both value that accrues directly to its participants and spillover value to society as a whole, going beyond individual mandates and control [42,49,50]. However, whole network studies, primarily hailing from the public and non-profit sectors, reveal a preoccupation with overall network effectiveness or societal value creation [41], and pay limited attention to partner value creation and the tension between societal value and partner value.

Solutions to large and complex social problems need engagement of the full array of actors from businesses, NGOs, and the government [7,10,42,46,51] to engage over an extended period of time to reach sufficient scale. However, given the long time horizon and the scope of change needed, it is unrealistic to expect organizations to contribute toward societal value creation unless they also deliver on their respective core goals [11,42]. For this to happen, we propose to combine the distinctive characteristics of networks and modular systems (see Table 1 below) to reconcile differing and often conflicting notions

of value between partners, facilitate smooth interactions, and ensure partner value as well as societal value creation. Such simultaneous creation of value to participants and value to society at large and the governance mechanisms that facilitates it has received limited attention.

We address this shortcoming as we develop our modular network governance framework.

**Table 1.** Characteristics of networks and modular systems.

| Characteristics | Networks (Whole Network) | Modular Systems |
| --- | --- | --- |
| Structure | Interconnected individuals/Orgs | Collection of individuals/Orgs |
| Basis for sub-groups/modules | Not defined | Expertise |
| Type of value emphasized | Societal value | Partner value |
| Locus of value creation | Network | Module |
| Dependence between actors | Varies from Low—High | None/Low |
| Integrating mechanism | Brokered; Shared | Market |

## 3. Toward a Modular Governance Architecture

We develop a governance framework that enables the following: first, manage the scale of the problem by mobilizing a variety of actors and breaking the problem down into a set of modules that each address a part of the overall problem by focusing on targeted outcomes; second, manage the modular interfaces to stitch components together to create a solution system; and third, simultaneously create value to participants and to society at large to make these collaborations sustainable and effective on sufficient scale. Thus, a modular network governance architecture enables a societal-scale solution by a unique system framework relying on flexibility and adaptability while accounting for the diverse and dynamic nature of the solution components.

### 3.1. Modularizing Societal Problems

Modularity is an intuitively appealing concept to break down large problems into independent and manageable modules and stitch them together to build a comprehensive solution for better convergence between economic growth and societal well-being. However, as noted, modularity has been principally applied to technological systems [32,52,53] rather than social systems. Thus, when we extend the tenets of modularity to a social setting, several challenges arise.

The first challenge is in grasping the boundaries of the system and creating the modules. Technological systems typically start as an integrated system, and the independent parts are modularized over time [54,55]. The overall system is well understood, that is, the role of each module and how modules fit together to function as an integrated whole is known. This allows the market forces of supply and demand to operate between modules and bind them. On the other hand, solutions to complex social problems have never functioned as integrated systems, and the way modules may be partitioned to facilitate the forces of supply and demand for solution components to operate is unknown. Thus far, actors in industrial and societal sectors—operating in their own silos, oftentimes driven by different ideologies, and being accountable to their respective set of stakeholders [56,57]—have addressed only a subset of the overall problem: that part that aligns with their core goals and capabilities. As a result, the individual contributions of these actors are isolated, incomplete, and disconnected [7,8], and an overall solution system remains elusive.

In order to unleash the potential of modularity on social systems, it is necessary to both get a good grasp of the overall problem and take as entry points any opportunity arising from strategic capabilities and competencies that a set of willing actors can bring. Unfortunately, though several actors may be independently engaged in addressing some facets of the problem, they do not have the time, motivation, or the capability to develop

an understanding of the overall problem [7]. This is because they are preoccupied with meeting their stated core goals and mission, constrained by a lack of resources to expand beyond their current priorities, or entrenched in their dominant institutional logic [58,59] that does not allow them to see beyond their own world view. Given these challenges, it is unlikely that existing, isolated efforts will organically coalesce to create a comprehensive or optimal solution [60]. Further, some facets of the problem are easier to address than others, creating an unbalanced supply of owners and solutions for some issues while others may go unaddressed.

Therefore, understanding the larger problem and mobilizing and orchestrating a network to address it comprehensively requires the deliberate effort and capabilities of a "convener" [60,61], also referred to as the backbone support organization [7] or broker organization [62]. By virtue of its mandate, this convener needs to have an extensive network of ties to different actors across society. Beyond the existence of ties, the convening entity should be one that can draw attention to the problem and exhort action from actors tied to its causes or solutions in relation to their core resources and capabilities [44,63,64]. Finally, the convener also needs to have a reputation and credibility that can accord legitimacy to the initiative [20,44], bring more actors into the fold, and ensure all actors are informed and engaged [63].

Government organizations have been a natural choice to play this convening role since both economic growth and societal well-being fall within their mandate, and they have considerable authority over other societal actors [39,63]. They can employ a variety of tools ranging from persuasion and incentives to strictly enforced regulations to shape the behavior of others. Multilateral development agencies, such as the World Bank, and international organizations, such as the World Economic Forum, could also play this convening role in partnership with domestic governments [62]. They have legitimacy, are purpose-driven, and can mobilize public and private actors [44,62]. The convener can also be a consortium of organizations from different sectors or standard-setting organizations [33] that is expressly formed to address a specific issue. Such a consortium could be a particularly effective convener since it would have broad-based legitimacy, access to a wide network of actors, and a large knowledge base. Further, the convening consortium could adopt a nested structure [8,9] involving organizations at the global, regional, national, and local levels, depending on the scale at which the issue is being tackled.

The next question that arises is, how does the convener build consensus among a diverse set of actors and partition the problem space into modules? Social issues tend to be multifaceted, spilling over the traditional boundaries of organizations, sectors, and jurisdictions [6]. Their complexity implies that it is not possible for the convener or any other actor to have complete knowledge of the problem [65]. They are also 'evaluative' [2], i.e., actors from different sectors and institutional settings tend to think about them differently, use different approaches to tackle them, and measure their impact [33,42,66,67]. This drawback means there is no complete or objective view of the problem upfront; rather, the contours of the problem emerge through communication, dialogue [68], and synthesis of different views and approaches. Therefore, to grasp the problem, the convener needs to forge ties and engage with a wide range of actors in the public, private and non-profit sectors [61], gathering and synthesizing multiple perspectives. Engaging with each new actor will bring forth new knowledge, capabilities, and constraints. Then, the convener needs to reconcile this new knowledge with what is already known and create a shared understanding of the problem among all the actors in the network. In essence, the convener acts as a bridge between different societal actors, facilitating a dialogue, reconciling different perspectives, and nudging the network of actors toward a shared understanding of the problem.

Once there is a shared understanding of the broader problem that needs to be addressed, it needs to be broken down into modules, i.e., smaller, manageable problems which one or more actors may address. Breaking broad problems into modules poses a few challenges in aligning actors and is core to the emergent and dialectic nature of modularization.

First, while trying to address one problem, new problems may be uncovered. This is because the constituents and boundaries of complex problems tend to be indeterminate [4]. Second, it may not be possible to determine, a priori, all the actors who need to be engaged in a module. This is because problems are deeply embedded in specific institutional contexts [67], and some aspects come to the surface only when an attempt is made to address them. Finally, there could be external shocks in the form of technology or policy changes that might require rethinking certain modules. Inability to recognize and cope with these dynamics can lead to the failure of these efforts [66]. What this means is that modularization cannot be viewed as a one-time activity. While the convener, in dialogue with various actors in the network, can partition the overall problem into modules, the solution network needs to evolve by inducting new actors and modules as necessary. In other words, new modules may appear, other modules may be transformed as the solution space evolves, and some modules may eventually disappear. Figure 1 illustrates the evolving nature of a modular network, which is a network of modules in which each module could itself be a network. In sum, the modular network structure is constantly evolving.

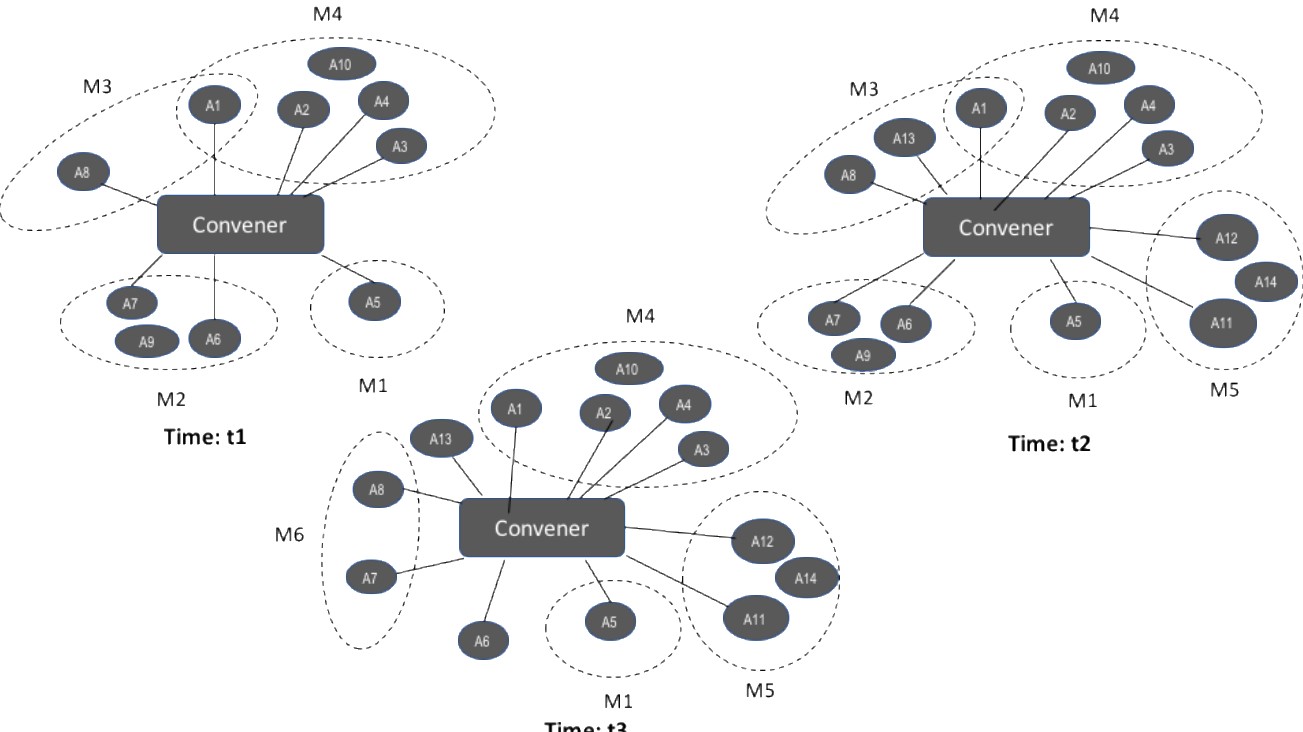

**Figure 1.** Illustration of modularization over time. Legend: A—Actor; M—Module; t—Time period.

To make the idea more concrete, consider the example of the problem of non-communicable diseases (NCD), due to unhealthy food consumption. One way of breaking this down into sub-problems and associated solution modules are indicated in Table 2. However, it is quite possible that in certain contexts, awareness about the linkage between high fat/high sugar foods and NCD is low. In such a case, this lack of awareness is a part of the problem and needs a separate module to address the issue. Furthermore, the same module may be structured differently in different contexts. Take the case of developing healthy food innovations. In contexts where the incidence of NCD is higher among rural populations, the focus may be on incorporating healthy agricultural commodities into the farming schedule. This will require participation from rural grassroots organizations as well as the agriculture department of the local government. In other contexts, where the incidence of NCD is high in urban areas, the focus may be on improving the nutritional profile of popular manufactured food categories. This will require participation from large food companies and distributors. In some other contexts, both these approaches may be

necessary. However, these contextual idiosyncrasies may not be known at the outset, and, therefore, solutions must emerge bottom-up from actors who have an intimate knowledge and stake in the context. In other words, the partitioning of the problem and the resulting modules need to be emergent and coalesce over time. Figure 1 and the example in Table 2 together illustrate how modularization plays out over time.

**Table 2.** Example of problem partitioning and modularization (for illustration only).

| Problem | Sub-Problems | Modules | Actors |
|---|---|---|---|
| To address the rising incidence of NCD due to unhealthy food consumption | Lack of awareness on nutritive content of food<br><br>Unavailability of healthy snacking/meal options<br><br>Lack of incentives to buy healthy food<br><br>Inadequate supply of health food commodities | • Research on food commodities and their health impact<br>• Easy-to-understand labeling<br>• Category marketing to build broad-based awareness of healthy food commodities<br>• Encourage healthy food innovations within large food corporations<br>• Incubation support for companies developing healthy food products<br>• Food-based prescription for NCDs incentives to buy healthy foods (e.g., Coupon scheme)<br>• Incentives to cultivate healthy food commodities (e.g., guaranteed minimum support price) | • Hospitals<br>• Nutrition association<br>• Public health department<br>• Food companies<br>• NGOs working with farming communities<br>• Agri research institutes |

Building on the preceding arguments and example, we posit that modularizing a societal problem needs the deliberate effort of a convener who uses a dialectic process to build an extensive network and create a shared definition or understanding of the problem. We also argue that modularization requires an emergent approach, one that allows a fluid modular structure. We formalize these propositions as follows:

**Proposition 1.** *Successful modularization of a societal problem (partitioning the problem into actionable modules) is associated with the following:*

*(a)   A convener who uses a dialectic process to assemble a network of societal actors and develop a shared understanding of the problem;*

*(b)   An emergent approach to modularization, allowing for a flexible modular structure.*

### 3.2. Modular Interface Governance

Another challenge in extending modularity to social systems is that they are imperfectly modular. In technological systems, modules are separated by a standard piece of hardware (e.g., USB) or data format (e.g., XML). Each module has to produce an output that is compatible with that hardware or data format, and by the same token, should be capable of taking inputs from that hardware or data format. Once defined, this creates complete independence between modules as long as they adhere to the agreed standards of input and output and partake in system-level outcomes. Such perfect modularity is difficult to achieve in social systems for several reasons.

First, the nature of inputs and outputs are often not inanimate but deal with human perception and behavior and complex interpersonal and interorganizational dynamics. For example, in the system to battle unhealthy food consumption, the module 'category' marketing to drive awareness of healthy food 'products' seeks to create broad-based awareness and drive demand for healthy food products. Essentially, it is aspiring to bring about a change in consumer buying behavior. Not only is health one among many motives driving choice, but contexts in which this occurs also matter. The output of this module (i.e., demand from modified behavior) is, therefore, not deterministic and can vary substantially, greatly affecting the modules that interface with it.

Second, modularity is primarily a principle or tool to simplify a complex problem into manageable components with as few interdependencies as possible between them while enabling their respective contribution to system-level outcomes. However, in social systems, this feature of modularity can also be leveraged to minimize conflicts by separating the conflicting parties into different modules [21,30]. This is particularly useful when trying to mobilize action from different societal sectors to address a common problem. Though the actors may agree to come together on a common platform to address the issue, some may have reservations about working directly together, due to a strained past relationship or irreconcilable ideological differences. In these situations, utilizing modularity can keep these actors apart and yet leverage their capability to work toward a solution without compromising the overall objective. However, such an arrangement is not ideal in terms of the design principles of modularity and may lead to residual interdependence between modules.

A third reason for the imperfect modularity of social systems is their emergent nature. As we argued earlier, the scope and boundary of complex social problems are indeterminate and emergent. Therefore, new issues may be discovered over time. It may not always be possible to integrate these new issues into existing modules with interdependencies. The older modules may not be able to address the new issue or have progressed to a point where broadening the scope would be counterproductive. New modules need to be put in place at such times, creating dynamic interdependencies between modules, possibly changing both project goals and partners as new solutions are needed.

This imperfect modularity of social systems implies that there is some residual interdependence between modules, requiring active management of the interfaces between modules. We refer to this as 'modular interface governance'—the set of coordination and monitoring mechanisms between two or more modules. Beyond imperfect modularity, there is another important reason why the modular interfaces of systems addressing social issues need to be actively governed. The market forces drive a modular system in the technology space. There are modules on both the supply and demand sides all along the value chain, with multiple players competing to fulfill the supply/demand needs. The competitive forces ensure that supply and demand meet, and the market clears. In the context of systems that are trying to address social issues, markets either do not operate efficiently or may not exist at all for many solution components. As targeted problem spaces are defined and modularized, markets are consciously created where they do not exist by designing modules to create both supply and demand. Take, for instance, the modules 'healthy food innovations within large corporations' and 'category marketing to build awareness about healthy food 'products' (Table 2). These are designed to respectively address the supply and demand side of healthy food consumption. However, since the demand is non-existent or nascent, competitive forces in favor of healthy foods are weak. Simulation studies have shown that, in such situations, purposeful effort to simultaneously move both supply and demand to a new state of being is required, in the absence of which market creation will fail [69]. In order to make this happen, interface governance is necessary.

The interface of a module with other modules depends on the type of interdependence that operates between them. After all, it is the presence of interdependence that necessitates interface governance to begin with. However, the type of interdependence may

vary. There could be sequential interdependence, which means that the output from one module forms the input for another. Interdependence could also be reciprocal, similar to sequential interdependence but operates in both directions [70]. There is a need for regular communication and close coordination if there is sequential or reciprocal interdependence between modules. Any delay or unforeseen challenges could prove costly to both modules. Further, by being closely engaged, modules may effectively deal with unfolding events promptly. Given the need for close coordination and the importance of relevant knowledge to deal with emerging situations, a shared governance mechanism where the concerned modules come together to govern the modular interface is suitable.

Another type of interdependence that could exist between modules is pooled interdependence [70]. Here, modules are not dependent on one another for inputs and work independently to produce outputs. However, their outputs aggregate to produce an output that is greater than the sum of individual outputs. The effect comes from having sub-parts that offer synergies and indirect benefits to the system. In the case of pooled interdependence, the governance mechanism needs to ensure that all the modules are on track. This is an administrative role and does not require intimate knowledge of the workings of the module. Additionally, pooled interdependence may exist between many modules because the more comprehensive the ecosystem to address the problem, the higher the number of modules that need to coalesce. Consequently, the governance overhead might be quite high and will require a dedicated entity to discharge the function. Given this, an NAO brokered governance will be appropriate for interfaces with pooled interdependence. We sum up the preceding arguments into these propositions:

**Proposition 2a.** *Modules with sequential/reciprocal interdependence between them are associated with shared interface governance;*

**Proposition 2b.** *Modules with pooled interdependence between them are associated with NAO brokered interface governance.*

### 3.3. A Modular Network for Simultaneous Partner and Societal Value Creation

Modules may be focused on developing new technologies, products, or services; creating new markets; building a supportive institutional framework; seeding behavioral change among consumers, and so on. Invoking the example of NCDs due to unhealthy eating, we can envisage modules as varied as researching the potential of various food commodities to decrease certain NCDs; rolling out a system of food-based medical prescriptions; seeding a pipeline of healthy food innovations within firms; designing incentives to promote healthy food consumption; ensuring affordable access to appealing nutrition food in vulnerable communities, and so on.

In such a modular network, each module addresses a specific facet of the overall problem and may be undertaken by a single actor or might involve a collaborative effort between two or more actors [71,72]. Suppose a module is implemented by a single organization. In that case, the governance of the module is straightforward and will be dictated by the nature and hierarchy of that organization, whether that be for-profit businesses, NGOs, or governmental agencies. For instance, a for-profit food company may commit to shift the bulk of its commercial food products and marketing toward more nutrition; a school district or a sport community organization may decide to ban sweet and fat food in its food service; government can adopt financial dis/incentives for un/healthy food formulation, marketing or consumption. However, given the scope of activities necessary to build lasting societal-scale solutions, many modules are likely to be interorganizational networks and will often cut across industrial and/or societal sectors and thus, need collective engagement [11,20,29,39,46]. The governance of these individual modules, which are interorganizational networks, depends on the structural and relational characteristics of the network and the external environment in which they operate [26,40,62].

However, one important aspect of these modules—the types of value they create—has received limited attention in the network governance literature. As mentioned earlier,

networks may create partner value for the organizations participating in the network (whether that be economic or fulfilling a social mandate), a broader societal value that accrues to society as a whole or both. The direct benefits provided by the network that meet the respective mandates of the participating actors are known as partner value. These could include benefits such as obtaining profit, deeper and broader engagement of vulnerable communities, and a healthier population. These are benefits beyond what the actors could achieve individually [46,48]. Networks may also aspire to create value for society at large due to the collaborative network [46,47]. This societal value may accrue to all individuals, communities, organizations, or institutions forming society, enhancing either or both economic and social well-being [50]. Suppose networks that aspire to address a social problem are to be sustainable and reach impact at scale in addition to providing societal value. In that case, they also need to create partner value prized by each of the participating actors. As obvious as this may seem, our society is not naturally oriented to achieve these twin goals simultaneously.

Generally, actors operate in silos, be these defined by industrial or societal sectors. Among societal sectors, creating societal value has typically rested on the government and civil society. On the other hand, the private sector often considers social issues to be diversions from their core mission of wealth creation and profit [73]. This does not entail the complete absence of the business sector in advancing societal well-being, but efforts remain clearly insufficient. Some firms earmark, for example, resources for social causes under the banner of corporate social responsibility [74] or corporate community involvement [75]. Others are more proactive, looking to engage the marginalized by creating value products for the base of the pyramid [76,77] or by integrating them into supply chains [78,79]. Still, others go further by creating shared value for the company and for the local community in which they operate [80]. Companies are also starting to report on their triple bottom line, making people and the planet impact key performance indicators alongside profit [81], and a few are taking the next step to formalize this broader mandate by getting certified as B corps [82]. Even so, these are exceptions rather than the rule, and the role of businesses in societal transformation remains primarily at the level of symbolic engagement, without engaging in a deep transformation of their core wealth creation activity.

Challenges lie in businesses' focus on economic interests and fears by governments and non-profit bodies of being tainted by those interests [18]. Public–private partnerships are routinely deployed for infrastructure development, provisioning of public services, and various other purposes [42]. However, there is a perception that the private actor is looking to maximize benefits for private wealth creation when engaging in domains tied to societal well-being, and there is little concern about the overall contribution to a better convergence between economic and societal well-being [40]. Therefore, governmental and international organizations often eschew potential conflicts of interest in their policy making and agenda setting. While avoiding conflicts in policy and agenda setting is commendable, it is impractical to expect concrete outcomes at scale in modern society without the participation of all sectors, including the private sector. A case in point is food and diet–related social problems, such as food insecurity or obesity. For governments and civil society to expect solutions at scale in promoting healthy food and diet without engagement with the private sector is counterproductive, especially when almost all of the food consumed in western countries, and a large percentage in developing countries, is procured from commercial sources [83,84].

Overcoming siloed structure and operations requires careful consideration of value created within and across modules for participating organizations and society. At the module level, it does not appear easy to create both types of value. This is because, by definition, a module deals with a sliver of the broader problem, has a clear and focused objective, and brings together a small network of actors to achieve that objective. Therefore, it may create societal value and/or partner value for a subset of the participating actors. However, expanding the module's scope to deliver societal value and create value for all the module participants can diffuse the module's goal and defeat the purpose of deploying

a modular approach. Thus, there is an inherent tradeoff between the effectiveness of a module and the breadth of value it creates. At the same time, modularity provides us the flexibility to spread the different types of value derived by the participating actors within and across the different modules to singly and collectively contribute to system-level outcomes.

We posit that each module may create either partner value directly to the participating actors or societal value, albeit with some spinoff of the other type of value. However, at the aggregate level, they feed off one another [47] to create both forms of value. In other words, the synergy comes from combining resources and capabilities [47] not just within, but across modules. For instance, continuing the food and NCD example, consider food companies involved in a module that creates an awareness campaign about certain healthy food commodities. The focus is clearly on creating societal value, and there is no direct value accruing to the firms from participating in this module. However, if the firms also develop food products based on those commodities, they can leverage the increasing consumer awareness to reap economic value. Additionally, the firms would likely engage in the former module only if there is a long-term potential of realizing value for themselves from the effort.

Building on the need for networks to create both partner value and societal value, we argue that though an individual module might focus on either value, at the systemic level, modules can come together to create both. Since modules address different facets of the problem, they will operate at varying time scales. Some may accomplish their goals and create value over a short time horizon, while others may come to fruition over a longer time horizon. However, so long as the system accommodates the creation of partner value for all actors in the network and provides a roadmap to accomplish the same, it is possible to mobilize and sustain a modular network.

The notion of partner and societal value creation has important implications for the governance of individual modules. In modules that focus on generating societal value, a key concern of governance is ensuring commitment and continued engagement from each participating actor in a time bound and transparent manner. This critical engagement poses a challenge because the social nature of the primary goal may limit the amount and quality of investment that each actor is willing to make and lead to problems of free-riding, especially from the private sector, leading to sub-optimal outcomes. Even in the absence of free-riding, modules creating societal value may often involve actors operating in different jurisdictions or geographical scales (e.g., community, city, state) with asymmetric resource and power positions. Such heterogeneity in the composition of the module makes alignment and cooperation challenging. Another key function of governance is to ensure that the value created becomes available to the society at large as it is intended and is not appropriated by a small set of actors. This might happen if the participating actors intentionally or unintentionally fail to share outputs widely. Consequently, this implies that significant external facing activity is necessary to ensure that only intended beneficiaries appropriate the value and to prevent deviant behaviors among participating actors. Given the sizeable governance function, an NAO form of governance is suitable. A dedicated, neutral broker can help overcome these challenges by scrutinizing each actor in the module and holding them to their commitments by continuous tracking. The broker can also actively engage with various external constituents to ensure that the value reaches the intended audience.

In modules that focus on partner value creation, governance needs to define what resources each actor brings to the collaboration and what value it would derive from it. Once this is defined, it needs to be monitored to ensure that all actors deliver on their promise and that there are no free riders. The value appropriated should be commensurate to the effort put in. Since the actors are driven by their own mandates, a shared governance mechanism, where the actors collectively govern the network, is suitable. Further, each participating actor may not be willing to cede control to a third party, no matter how

neutral, and prefer a shared governance mechanism through which they can control how value is created and appropriated.

We recognize that a module could create either or both partner and societal value by design or as an unintended byproduct. Therefore, our representation is stylized and captures the emphasis on one type of value over the other, rather than a complete absence of one. We also assume that the actors engaged in a module are equal partners in the collaboration. We formalize our arguments through these propositions:

**Proposition 3a.** *Ceteris paribus, NAO governance is more effective than other forms of governance for a module creating societal value.*

**Proposition 3b.** *Ceteris paribus, shared governance is more effective than other forms of governance for a module creating partner value.*

## 4. Discussion

We have developed a governance framework to build and manage a multi-actor, multi-sector platform that aims to address societal challenges in a lasting manner and at a sufficient scale. The governance framework advanced in this paper makes several important contributions. First, it provides a theoretically grounded actionable framework for addressing complex social problems. Second, it extends modularity into the social sphere. Third, it paves the way for discussing how society can harmonize the engines of wealth creation and societal well-being.

### 4.1. An Actionable Approach to Complex Social Problems

We advance the conversation on addressing pressing, complex social problems in two ways. First, the modularity-based governance advanced here provides a theoretically grounded framework for a general "divide and conquer" approach that is put forth to address complex social problems by earlier work. For instance, convergent innovation [11] proposes stitching project portfolios that encompass technical, business, social, and institutional innovation, targeting both supply and demand. The collective impact framework [7,8] proposes creating multiple working groups to implement an overall strategy agreed upon by participating organizations, with a recent extension promoting a multi-layered backbone structure aligning a common vision and goals across jurisdictions [9]. Our modular network governance framework provides a theoretically grounded framework that can be adapted to any of these emerging approaches to address social problems.

Second, the literature is divided on whether a top-down or a bottom-up approach is more suitable to address complex social problems. For decades, most assumed that a top-down approach was the best strategy to solve these problems because of its integrated and comprehensive nature. However, a bottom-up experimental approach has found favor [85], based primarily on realizing that a central entity cannot drive the solution to a complex, contextually embedded, emergent problem. Our framework resolves this dichotomy by arguing that a convener is needed to catalyze and facilitate solutions by bringing together actors and positioning their respective interests, capabilities, and actions with those of others. However, to reach sufficient scale, both the problem scope and the solution itself need to emerge bottom-up to create ownership and an ecosystem for change [86]. Together, our framework provides important guidelines for practitioners and policy makers looking for pragmatic solutions to pressing social problems.

While the framework proposed here is a useful starting point to address complex social problems, the immediate next step would be to undertake empirical work to validate and refine it. There are two ways to pursue this. The first is to undertake inductive research based one or more case studies that have sought to address a complex social problem. For example, in the developed countries, this may be related to universal affordable healthcare or the problem of NCD. In the developing countries, these may be around undernutrition, basic education or water and sanitation. Semi-structured interviews with the convener and

the actors involved in such projects can reveal if and how the problem was modularized and how the various modules were tied together. A second and more interesting approach would be to undertake action research and follow a project as it unfolds. This can generate rich data on how the problem is framed, how societal actors are co-opted into the solution space, the role of the convener, how the problem/solution gets reframed and modularized over time, and the issues that surface over the course of the project. This would be a multi-year, longitudinal study that can reveal important micro processes and refine the framework advanced here.

### 4.2. Modularity in the Social Context

Our framework opens several new lines of inquiry on the scope, limits, and dynamics of modular social systems. For instance, we have only begun the scholarly conversation on modularization and interface governance, the central considerations in creating effective modular systems. We have proposed that modularization is a dialectic, emergent process. While this is a useful starting point, research based on real projects is necessary to validate and refine what is proposed here, and further tease out the factors that influence modularization. Furthermore, we have limited our theorization on interface governance to focus on the role of interdependence between modules since it is a direct consequence of imperfect modularity. Future research could explore other contingencies (e.g., module characteristics, external environment) that influence interface governance.

Further, the need to orchestrate the modular interfaces creates the need for an administrative layer. In order to leverage the full benefit of modularity, however, it is desirable to keep the administrative overheads of the system at a minimum. The question then is as follows: how can this be achieved? How do we determine modular boundaries? Can the modules be created so that a majority of them can operate within hierarchies or use market mechanisms? What other considerations should go into the design of modules? Many other research questions arise as we seek to understand the potential and limits of modularity as a means to combine individual and system-level action for addressing complex social problems. What are the challenges of modular social systems, and how can they be mitigated? How do modular social systems evolve over time? Do modular social systems go through different stages? What is the role of the convener over these different stages? These and several other questions promise to enrich the scholarly conversation on modularity as well as the conversation on how modularity can help address social problems. Empirical test beds are available for observation, as a major effort has been made worldwide to address sustainable development goals. These also provide the opportunity to experiment to bring forth scientifically sound and solution-oriented goals.

### 4.3. Bridging Partner and Societal Value Creation

Since the onset of the first industrial revolution, businesses have pursued economic value creation anchored in commercial exchanges between them and their customers (be they consumer or business-to-business clients) largely without consideration of the negative externalities progressively created for the natural environment or for human development and health [10]. Any attempts to constrain these costs and to create societal value have been, at best, an afterthought and aim mostly to "fix" problems created by single-minded, unbridled private economic value creation. However, suppose that we are to move in the direction of sustainable development. In that case, we must bridge the engines of economic and societal well-being and ensure that these twin values guide the actions of each actor in society. This is an aspirational goal that requires substantial behavioral change at the individual, organizational and societal levels.

Taking a step in this direction, we look at how the solution system to a complex problem can simultaneously create value for participants and society at large, thereby making it sustainable and effective at scale. We extend the network literature that, to date, assumes that interorganizational networks focus on a unitary goal: either value creation for participating organizations or value accruing to society as a whole. By marrying this

literature with modularity, we conceptualize the solution space as a modular network, where each module is itself a network, but with minimum interdependence with other modules. While some modules may primarily create societal value, others create value for participating organizations, and a limited subset may create both. Thus, the modular network creates value for participating actors and broader societal value, providing a way to bridge the deep divide between investment in economic and societal well-being that has characterized modern society.

This is a small first step toward articulating how to build sustainable collaborative systems that create value for every actor involved while also furthering overall economic and societal well-being. Much empirical research is required, such as case studies or action research projects, to test and strengthen the proposed framework. The context of this paper is to redress existing social problems. The next frontier would be to understand how ecosystems can be created in a viable and resilient manner to progress the engines of economic growth and societal well-being in lockstep and avoid the negative externalities of economic growth that have created the issues we are battling today.

*4.4. Can Digital Platforms Serve as Conveners and Modular Ecosystem Builders*

Through significant digitalization advancements throughout the last century, most aspects of individual and organizational life are surrounded by digital technologies [87,88]. As digital technologies and infrastructures have evolved from a set of task-specific tools into a transformative contextual force that enables radical reorganization of economic and societal value creation activities [88–90], it is now recognized that digitalization shapes the dynamic of value creation and appropriation [91].

Now the question is as follows: given the rise of digital platforms, what role could they play in modular network governance? We anticipate that digital platforms could facilitate modular network governance in three ways. First, a digital platform could act as a co-convenor that allows different societal actors to connect through the platform, exchange information, and facilitate communication, i.e., in building the modular network platform. Second, it can facilitate interface governance by managing interdependencies between modules and flagging any delays that could derail the project. Finally, a digital platform can also be an output of the modular network activities, accumulating knowledge and capabilities that deliver societal value and partner value over time. Further research is needed to explore the role of digital platforms in modular network governance and how they interface with the complex and dynamic inter-organizational mechanisms.

*4.5. Boundary Conditions and Limits to Modular Network Governance*

It is important to understand the conditions under which modular solutions to complex social problems work well and the limitations to their applicability. First, the initial conditions in the action arena (local, state-level, national, or global) need to be favorable. This means there needs to be some degree of awareness and public discourse about the problem and a desire to seek progressive solutions that can mobilize concerned actors towards action. For instance, the use of standards, such as fair trade or efforts to encourage consumption of local farming, can be important signaling mechanisms to transform the agri-food value chains. A critical mass of large actors across different sectors also needs to be interested and willing to engage with the problem. Engagement is critical to ensure that the power equations are equalized across societal sectors, diverse interests are represented, and a small set of powerful actors does not hijack the agenda.

Second, the problem must warrant a modular approach. Modular network governance comes with a substantial overhead in the form of governance and coordination of many actors. Therefore, one would only select a modular approach where the number of actors is significant and the dynamics of their interaction sufficiently complex to render other approaches ineffective. Third, the problem needs to allow opportunities for both partner and societal value creation. In other words, it should have the potential to cement convergence between economic and societal well-being.

Finally, this is a small contribution toward addressing large societal problems. By no means is the proposed governance framework comprehensive: it is a starting point. For instance, we limit our theorizing to the modularization of a complex social problem. However, modularizing a problem is only the beginning. Given the diversity of actors involved, and their varied goals, motivations and world views, the ongoing orchestration of such a modular network poses many challenges. We do not dive into the social dynamics that could play out over time and their implication for governance. We also recognize that we have woven together theoretical streams from two different fields—technology management and social networks—to develop this framework. We rely on scholars from these fields to enrich this framework.

### 4.6. Conclusions

The modular interorganizational network governance framework we present here adds scientific support from the management discipline to a societal scale solution to address complex social problems and grand challenges, as encapsulated in the 17 Sustainable Development Goals [92] to be achieved by 2030. Our framework provides a concrete approach toward turning these goals into reality.

**Author Contributions:** Conceptualization, S.K.J., E.R.G. and L.D.; writing—original draft preparation, S.K.J.; writing—review and editing, S.K.J., E.R.G. and L.D.; visualization, S.K.J.; supervision, L.D.; funding acquisition, L.D. All authors have read and agreed to the published version of the manuscript.

**Funding:** This work was primarily supported by post-doctoral fellowship to Srivardhini Jha from the CGIAR Fund through the CGIAR Research Program on Agriculture for Nutrition and Health; Complementary funding came from International Development Research Centre (IDRC) [grant numbers 107400-006, 2015]; Fonds de recherche du Québec—Société et culture (FRQSC) [grant numbers 2015-SE-179342, 2014]; and Social Sciences and Humanities Research Council (SSHRC) [grant numbers 410-2010- 2258, 2010].

**Institutional Review Board Statement:** Not applicable.

**Informed Consent Statement:** Not applicable.

**Data Availability Statement:** Not applicable.

**Acknowledgments:** The support of the following research assistants at different stages of the research and paper writing is also gratefully acknowledged: Claire Burgoyne, Cameron McRae, and Hassan Ebrahimi.

**Conflicts of Interest:** The authors declare no conflict of interest.

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
