# Peer review of "Modular Interorganizational Network Governance: A Conceptual Framework for Addressing Complex Social Problems"

_sustainability, doi:10.3390/su131810292_

Round 1

Reviewer 1 Report

Although the article presents only conceptual framework, its topic is scientifically interesting and has great potential. It is relevant to the Sustainability journal theme and scope (section: Economic and Business Aspects of Sustainability). I also believe that the article suits for the journal special issue “Technological and Organizational Innovation for Sustainable Development”. In my view, a few issues may be improved to increase the impact and attract more readers. They are listed below.

My main objection concerns the attempt to fit the imperfect modularity of social systems within the typical modular system that is applied mainly to technological systems. In particular the typical approach do not take into account interdependencies between different actors and problems. The authors are aware of this problem and discuss it extensively. However, some general doubts about the validity of this approach still remain.

The journal has no word limits, however in some places the authors' argument seems to be too diligent and multi-threaded. This can make text unclear to a reader who loses the story while reading. Therefore I suggest introducing some shortcuts and making it more compact (eg. Section 3,1.). However it is not a must.

In the article the authors have formulated three propositions. In the light of the provided argume nts they sound convincing. However, as the potential improvement of the third proposition (both 3a and 3b) I suggest to clearly state from what NAO governance and shared governance is “more effective than”. Although it is explained in the proceeding text, in the section summary it is not clear.

In addition, the empirical contributions of the research may be explained in detail (i.e. how to use the proposed framework in practice).

Author Response

Dear reviewer,

Thank you for time and constructive feedback in the paper. Pls find attached a point-by-point response to your comments in the attached file.

regards

Reviewer 2 Report

  1. The article stresses on social decision making process, seen mainly from a mechanical or technical point of view, by using modules or splitting decision into modules. The solution the authors develop tends to act against the purpose it follows by disregarding the essential role of social actors. These are organizations and/or individuals making decisions in a particular institutional framework designed by rules and regulations which influence both the decisions that social actors made and the outcomes that the decision makers want to implement. See North, D. C. – Institutions, Journal of Economic Perspectives, Volume 5, Number 1, Winter 1991, pages 97-112; North, D. C. - Institutions, Institutional Change and Economic Performance, Cambridge: Cambridge University Press, 1990. Although the authors emphasize the need to avoid the mere transfer of the technical solution of modulation to social decision making process, they tend also to embrace it during the entire article.
  2. Authors’ initiative to review the literature on the governance of modular systems and emphasizing its faults is to be appreciated. Also, observing the superficial approach in this matter (i. e. a module is completely under the control of a single organization and each module adheres to the interface specifications) is a very important contribution of the authors. However, the mere emphasis of these particular problems and even providing a solution to tackle them do not solve the issue in question: how social objectives, problems or potential solutions can be managed through social organizations, whatever the „technical” manner to do it. This particular question becomes more relevant as the authors see governments and international organizations such as World Bank or World Economic Forum as main actors in the process of the governance of modular systems. See Buchanan, J., Tullock, G – The Calculus of Consent: Logical Foundations of Constitutional Democracy, University of Michigan Press,  Buchanan, J.  M. - The Limits of Liberty: Between Anarchy And Leviathan. Chicago: The University of Chicago Press, 1975.
  3. The solution that the authors develop in this article does not make a considerable improvement in tackling some of the deficiencies they emphasized in the article. One possible explanation for this is that the authors provide also a mostly „technical” solution to a social matter. It may be useful to see the whole process of social decision making and outcomes as a less abstract model and accepting that such „model” includes actors that have purposes, own goals and act under a given and dynamic set of rules and regulations. The actors are organizations and individuals and even if they are mobilized in order to break the social problem into modules, they still can shift between various options and outcomes due to their own/organization interests. Another problem with the solution that the authors are developing in this article is that an organization, having the legitimacy and being purpose-driven as main assumptions, would face a major problem when trying to address social problems: gathering the relevant knowledge concerning that particular social problem. See Hayek, F. A. – The Use of Knowledge in Society, The American Economic Review, Vol. 35, No. 4. (Sep., 1945), pp. 519-530, Stable URL: http://links.jstor.org/sici?sici=0002-8282%28194509%2935%3A4%3C519%3ATUOKIS%3E2.0.CO%3B2-1
  4. The authors correctly emphasize a series of difficulties associated with the solution they develop, but unfortunately they fail to explain how this particular solution tackles these problems. In fact, difficulties emphasized are inherent to all social cooperation processes, especially when it comes to making social decisions which involves many actors having various interests, not necessarily converging ones. It was expected that the authors’ proposed solution to solve these problems; but instead, the mere problems are presented as limitations of the proposed solution. Therefore the solution itself is not quite a response to the social decision making process, but merely a technical surrogate. It is also difficult to pretend having developed “a general model applicable to diverse approaches proposed to address complex problems”. This statement contradicts the previous statements of the authors regarding the limitations that accompany the model they developed. By its own nature and concerning the particularities of complex social decisions, it is hazardous to pretend that a “general model” could apply.
  5. In order to advance on research concerning the role of digitization, see Fanea-Ivanovici, M.; Muşetescu, R.-C.; Pană, M.-C.; Voicu, C. Fighting Corruption and Enhancing Tax Compliance through Digitization: Achieving Sustainable Development in Romania. Sustainability201911, 1480. https://doi.org/10.3390/su11051480
  6. How can be described/which are the characteristics of a „dedicated convener”? How can it be identified?

Author Response

Dear Reviewer,

Thank for your time and constructive feedback on the paper. Pls find a point-by-point response in the attached file. We hope this satisfactorily addresses the issues you have raised.

regards

Reviewer 3 Report

Thank you for this timely and thoughtful paper it was an enjoyable read.  I have made my suggestions below but i am willing to recommend that it is published in the journal. Though admittedly this is not my specific area  of academic expertise, i recognise the value of this work and the potential wide applicability or usefulness of the content here for society both within and beyond the parameters of academia. 

Your opening sentence is an accurate yet under cited large claim.   This would benefit from some citations as it is the foundation for the arguments that follow. Your second point would also benefit in the same way as you are assertive in your tone but don’t necessarily support your points as immediately as they could be - do consider this throughout as one persons 'obvious' is another persons 'prove it.'.  I am aware that this is a point of construction rather than content however so this remains a suggestion, not an instruction.

The writing style and tone is articulate and academic throughout, which is of a very high standard. However, with caution, at times this style of writing can manufacture accessibility issues -  it may be worth considering creating an accessible version/resource that is more accessible to more lay audiences as the subject matter is undoubtedly useful to various sectors both within and external of academia. Further, i am not suggesting a major overhaul, but there are times where the language used may be considered unnecessarily formalised/complex and actually simpler phrasing of the same points may lead to greater impact through wider readership. This is not a criticism of the wording - it  is very well written - just an advisory suggestion for if the authors want to contribute to the accessibility and wide reach of their paper. 

The phrase “incentive” could be problematic here - incentive can be a complex concept and heavily weighted with a variety of intersecting factors impacting it. Again, this is your choice but you may choose to define, unpack or change this word so it doesn’t carry as much of an antagonistic tone. (End of p7 to top of p8)

Figure 1 is very condensed and the small writing in white on a dark background is challenging to read for myself - and will be for others, potentially due to a range of potential eyesight challenges for example, but also just because of the nature of the presentation- could this be made clearer? writing emboldened? or even given its own page to resize for enhanced clarity? Not essential but from an accessibility perspective subtle changes would make a big difference. 

The discussion is detailed and useful and is a real strength of the piece - the limitations section does lean more towards the limitations in application of the model rather than limitations of the work itself so some more attention or detail could be given here but overall this paper makes a useful and valuable contribution and i am happy to recommend it to go forward for publication in the MDPI sustainability journal.

Author Response

(The authors gave the same response as above.)

Round 2

Reviewer 2 Report

The authors make considerable improvements by adding the suggested literature references and useful arguments for the theoretical framework they advanced in this article. It is to be appreciated the effort (a particular paragraph has been added in section 4.1) of further arguing on the complexity of social problem solving. The authors advance also a specific solution to validate their work, an empirical one based on inductive research. This suggests that the process of addressing complex social problems may be divided on two levels: "macro" and "micro". The article stresses on the "macro" level and further inductive research should offer more insights concerning the "micro" level, meaning that particular level of decision making and getting the social actors together. However, separating on "micro" and "macro" level is difficult and, at some point, useless. Putting social problems on the solving agenda is a result of particular interests that occur at a "micro” level. The main question in this matter is what social problems get to the solving list, rather than how it would be solved. Both questions, however, have a common answer: social, economic, political actors assuming the task of solving these problems and pursuing the presumptive benefits attached. When these actors involve government agencies, the solution needs "public choice”. This is the argument, in my opinion, for considering "public choice" literature and approaches as a main pillar in building (even) a theoretical framework for social decision making process.

However, the improvements to this article are satisfactory and my decision is to be published in this added form.